# Knowledge mobilisation: an ethnographic study of the influence of lay mindlines on eczema self-management in primary care in the UK

Fiona Cowdell[1,2]

¹Faculty of Health, Education and Life Sciences, Birmingham City University, Birmingham, UK
²Centre of Evidence Based Dermatology, University of Nottingham, Nottingham, UK

**Correspondence to**
Professor Fiona Cowdell;
fiona.cowdell@bcu.ac.uk

## ABSTRACT

**Objective** To investigate the way in which mindlines, 'collectively reinforced, internalised tacit guidelines', are constructed among lay people with eczema in primary care.

**Design** Ethnographic study.

**Setting** Observation in one general practice in the UK and interviews across central England.

**Participants** In observation, patients in the participating general practice regardless of presenting complaint and in interviews, people with eczema or parents of children with eczema (n=16).

**Results** Observation of over 250 hours and interview data were combined and analysed using an ethnographic approach through the lenses of mindlines and self-management. Four themes were identified: doctor knows best; not worth bothering the doctor; I need to manage this myself; and how I know what to do. Themes were set within the context of four broad typologies of lay people's approach to self-management: content to self-manage; content to accept practitioner management; self-managing by default; and those referred to secondary care.

**Conclusions** This study is the first to examine how lay eczema mindlines are developed and to recognise typologies of people with different need for, and receptiveness to, information. Lay eczema mindlines are constructed in many ways. The outstanding challenge is to find strategies to revise or modify these mindlines by adding reliable and useful knowledge and by erasing outdated or inaccurate information.

## INTRODUCTION

Knowledge mobilisation (KM) can, at its simplest, be defined as 'moving knowledge to where is can be most useful'[1]; it involves concerted efforts to create, share and use research and other forms of knowledge.[2] Many knowledge mobilisers recognise that knowledge sharing is relational,[3] constructed from social interaction[4] and context specific.[5] One potential KM strategy is influencing 'mindlines', a concept developed from extensive ethnographic work in primary care. Mindlines are 'collectively reinforced, internalised tacit guidelines' on which clinical

## Strengths and limitations of this study

► First ethnographic study to examine the development of lay eczema lay mindlines.
► Diverse sample patients and parents of children with eczema.
► Ethnographer was a lone researcher.
► Results may be context specific.

decisions are made.[6] Mindlines are informed by the work of Polyani[7] and Nonaka and Takeuchi,[8] who suggest that not all knowledge is conscious and explicit and that tacit knowledge, in the form of technical know-how and unconscious schemata, is a far more powerful influencer of action than formal codified knowledge. Gabbay and le May[6] suggest that mindlines are built on a flexible, embodied and intersubjective understanding of knowledge, which takes into account the local context and the existence of multiple realities. They represent a complex amalgamation of knowledge sources such as communication with colleagues and opinion leaders in the field and from personal tactic knowledge built up over time.[6]

A synthesis of 10 years of mindline literature (n=340) reports four key areas of study: nominal, in practice, theoretical or philosophical and solution focused.[9] Solution focused studies (n=28) actively promoted and supported the development of valid collective evidence-based mindlines. These researchers emphasise the importance of relationship building, collaborative learning and effective leadership.[9]

Gabbay and le May[6] hint at the existence of a patient equivalent of mindlines but do not develop this notion. Similarly, this possibility is poorly represented in other literature.[9] The term clientlines appears in one study[10] but is not fully explored. Repeating the search strategy employed for this synthesis reveals

no new studies on patient mindlines since the original review.

If primary care practitioner practice is influenced by mindlines, it seems reasonable to think that patients, particularly those who are self-managing long-term conditions, such as eczema, are likely to have a lay equivalent. The phrase 'lay mindlines' has been coined because many people who self-manage a long-term condition would not classify themselves as patients.

Atopic eczema (also known as atopic dermatitis and commonly referred to as eczema) is a long-term relapsing skin condition. It can cause untold suffering both physical and psychological and can have a detrimental impact on quality of life for individuals and their family.[11][12] It is one of the 50 most burdensome diseases globally.[13] Eczema affects around 1 in 5 children and 1 in 12 adults in the UK.[14] Eczema is treated in primary care in 97% of cases[15] and has a high self-management demand. The mainstay of treatment is regular and consistent application of topical medication,[16] predominantly emollients and steroid preparations. Treatment failure is common[17][18] and wastage of prescribed preparations high.[19] Leave-on emollients alone cost around £71 million per year in England,[20] and 'steroid phobia' is a common phenomenon.[19] Primary care consultations can be unsatisfactory for both patients and practitioners.[21][22]

Self-management of long-term conditions is a policy imperative.[23–25] There is no agreed definition of self-management[26]; it is broadly concerned with sustained efforts to maintain or improve health. Self-management is underpinned by interventions designed to increase capacity, confidence and efficacy of the person to perform the required activities.[27] The challenges of self-management of long-term skin conditions are well documented.[28] In eczema, supportive interventions have included self-management programmes[29] and educational and psychological interventions.[30–32] However, the active ingredients of these interventions are poorly understood, they have restricted availability, can be costly to provide and have variable impact.

Given the prevalence of eczema, the high self-management demand and the challenges of primary care consultations, it seems prudent to investigate the way in which mindlines are constructed among lay people with eczema in primary care. This understanding may be used to develop novel approaches to influencing these mindlines with the intention of mobilising relevant, accurate, up to date and contextually appropriate knowledge to enable people with eczema to self-manage as effectively as possible.

## METHODS
### Aim
To understand construction of lay eczema mindlines in primary care.

### Design
Following the lead of Gabbay *et al*[33] an ethnographic approach was employed. Ethnography is founded in anthropology and is concerned with the systematic study of people and cultures.[34] Data are collected through extensive observation with informal conversations, field notes and interviews.[35][36] Data were collected in one general practice for depth and in a large and super diverse geographical area for breadth.

### Setting, participants and process
All data were collected by the author, a researcher and registered nurse, between January and June 2017. The general practice was identified by a local clinical research network. It was a research and education active, urban general practice in central England, with a patient population of approximately 10 000. Prior to data collection, the researcher attended two practice meetings to introduce the study. Observations were collected over more than 250 hours. The researcher adopted the role of social-participant-as-observer,[37] which included activities such as cleaning couches and taking prescriptions to the local pharmacy. Patient journeys were observed from contact with receptionists, either face to face or on the telephone, through the waiting room and during telephone consultations with general practitioners (GPs). Further observations were completed during in-person consultations with GPs, GP trainees and locums and nurses. Baby clinics run by health visitors and interactions with staff in an associated pharmacy were also observed. Practitioners briefly introduced the researcher and the study to each patient, on occasion consultations were exited at the request of the patient, practitioner or of the researchers own volition. During observation, copious field notes were written; informal conversations were either written contemporaneously or audio-taped. Entire clinics were attended regardless of presenting complaint; this offered valuable understandings in the context of other long-term conditions.

Interviewees were recruited from two sources. First, invitation letters were sent to patients from the general practice who had a diagnosis of eczema recorded in their medical records and who had been prescribed emollients during the last year, indicating that their eczema may be a concern. Letters were sent to a group selected to represent the broad spectrum of patients including different age, gender and nationality. Second people were recruited via a higher education institute website with a reach of over 5000 people including staff, both academic and professional services, and external subscribers. Several participants were recruited by word of mouth from people who had seen the web recruitment information. Potential participants were sent an information sheet prior to meeting, rapport was established at the beginning of each interview. Single, semistructured interviews focusing on sources of knowledge were completed using a topic guide (box 1). Participants (n=16) were lay people registered at the observed practice (n=8) and lay people recruited

## Box 1  Patient and parent interview topic guide

► How long have you or your child had eczema?
► How bothersome is it?
► How often do you have to see a nurse or doctor about it?
► What treatments do you use most often?
  – Where do you get these from?
► How does your nurse or doctor choose a particular treatment?
► Do you understand why they recommend a particular treatment?
► How much do you and your doctor share the decision about what treatment to use?
► Where do you else do you get information about eczema?
  – Can you give any specific examples?
► Does your doctor or nurse refer you to any external sources of information?
  – How do you know if this information is reliable?
  – How could we best get this information to you and other patients or parent?

**Table 1** Demographic details of participants (two identifiers removed)

| Gender | Age |
|--------|-----|
| Male | 42 |
| Female | 24 |
| Male | 24 |
| Female | 27 |
| Female | 35 |
| Female | 38 |
| Female | 45 |
| Female | 38 |
| Male | 11 |
| Male | 7 |
| Male | 22 |
| Female | 32 |
| Female | 32 |
| Male | 59 |
| Female | 78 |
| Female | 17 |

via the website (n=8). Lay participants were all residents in the Midlands of the UK, which is recognised as being an area of superdiversity, defined as an area with citizens of multiple nationalities and socioeconomic status. The sample frame for interviews was people with medically diagnosed eczema. Sampling was necessarily pragmatic, but use of maximum variation purposive sampling ensured a mix of ages, gender and ethnic background (table 1). Interviews were conducted in the GP practice, people's own homes and the researcher's workplace according to personal preference. Child participants were accompanied by a parent. Each interview was audio-recorded and lasted from 25 min to 90 min. The aim was to understand how mindlines are constructed among lay people with eczema in primary care. Data sufficiency was

achieved when no new sources of knowledge were identified in interviews.

An iterative approach to data collection and analysis was used with initial findings being used to guide further data collection.[35][36] Audio data were professionally transcribed. The researcher then proof read transcripts against recordings for accuracy. Inductive data analysis was completed though the lenses of mindlines and self-management using the technique of Gabbay *et al*.[33] Transcripts and field notes were read in full to get a sense of the whole and then manually (using Post-it notes) coded, categorised and merged into themes. Following theme development, relevant sections of the data were reviewed to ensure authentic interpretation and use of participant language.

### Reflexivity
A reflexive stance was maintained throughout, acknowledging own subjectivity and positioning as a nurse and skin health researcher and the impact that this may have on the study.[38] An audit trail of decision making is provided throughout this paper to demonstrate the robustness of the study.[39]

### Patient and public involvement
Lay people were involved in the development of the research question and in planning the design of the study. Results will be disseminated to participants in the form of a brief summary.

## RESULTS
One unexpected observational finding was the low number of consultations specifically for eczema, despite 19.5% of the practice population having a diagnosis of some type of eczema on their computerised records. The majority of eczema consultations observed involved secondary concerns raised at baby clinics. The data presented below explain this phenomenon.

Four core themes were identified in the data: doctor knows best; not worth bothering the doctor; I need to manage this myself; and how I know what to do. These themes are set within the context of four inductively generated broad typologies of people's approaches to self-management: content to self-manage; content to accept practitioner management; self-managing by default; and those referred to secondary care (figure 1). The typologies are explained, and each theme is discussed with examples from the data below. Lay person typologies and themes are cross-cutting, although the focus in this paper is on the first three typologies as it is these groups who are wholly treated in primary care.

In observation and interviews, it was clear that participants had differing approaches to self-management that they could broadly be identified in four typologies. The 'content to self-manage' group predominantly consisted of people who had mild, relatively untroublesome disease that they managed on an ad hoc basis. Others had learnt

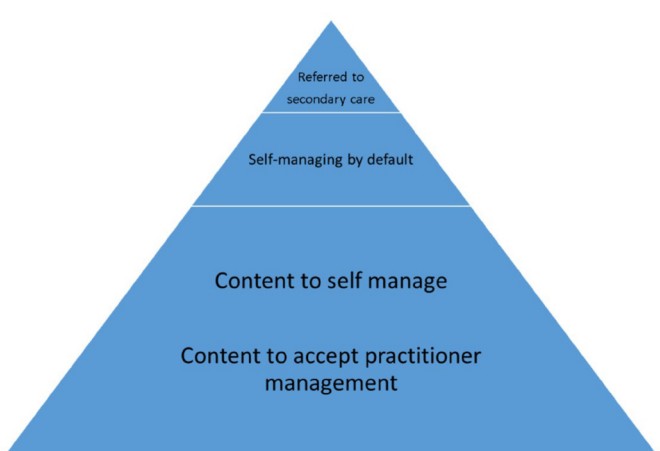

**Figure 1** Typologies of lay people's approach to self-management. The four broad typologies of lay people's approaches to self-management: content to self-manage; content to accept practitioner management; self-managing by default; and those referred to secondary care.

how to self-manage effectively, often through a process of trial and error and reported 'I know what works ………… I get the creams and I treat it'. The 'content to accept practitioner management' group showed relatively little inclination to be involved in making treatment decisions, for example, 'I'm not much of a researcher, I need someone to tell me'. Those 'self-managing by default' were the largest group, possibly as they were the most likely to want to be involved in this type of study. They reported repeated, often unsatisfactory consultations and ultimately minimising primary care consultations. For example, 'they chuck creams at you and when you finish them the problem comes back because they've not addressed the problem …. I don't tend to go back because I know what's going to happen'. The final typology of 'those referred to secondary care' is self-explanatory. The few participants who had been treated in secondary reported having an action plan, which was used as a basis for treatment decisions by themselves and in conjunction with the primary and secondary care practitioners.

### Theme 1: doctor knows best

A minority of lay people had little desire for more knowledge about their eczema. For some, this was because it was not bothersome and could readily be managed with use of emollients and occasional topical steroids. Generally, this group requested repeat prescriptions as required and booked a telephone appointment or consulted a GP if and when they needed topical steroids. They were satisfied with gaining access to any available practitioner believing that all the required information was available to GPs on their electronic records. Their knowledge was predominantly from practitioners. Eczema consultations with nurses were rare but reported as providing the most useful information particularly imparting 'the simple stuff' such as about *how* to use topical treatments. Some lay people had absolute faith in practitioner stating, for example, 'I think they should make the decisions, I

don't actually know' and 'a nurse or a doctor …. I would definitely believe them'. This group noticed when their repeat prescriptions for emollients had been changed but did not question this, assuming a sound rationale.

### Theme 2: not worth consulting

Three lay beliefs underpinned this theme: perceived difficulty in getting a GP appointment; eczema being viewed as a trivial condition; and primary care practitioners perceived to have little expertise in dermatology. The system for making appointments in the observed practice involved telephoning at the beginning of each session (08:00 and 12:30 each weekday); a few appointments were available each day but once they had been taken, patients were added to a call back list. During the telephone call, the GP would either deal with the problem or, where necessary, invite the person for a face-to-face consultation the same day. This system was inconvenient for some, particularly working people, and made it difficult to see the same GP over time. Variations on the challenges of getting a timely appointment with a specific GP were reported by all. People who had access to prebookable appointments had concerns about the length of time they had to wait to see their chosen practitioner. This diminished the doctor–patient connection in which 'relationship and trust is so important'. Having to explain their eczema journey each time they saw a different GP could lead to intense frustration at 'having to repeat history' each time and 'going full circle with doctors'.

Some lay people perceived that their eczema, while bothersome, was a trivial condition and ascribed the same perceptions to practitioners. This was supported by observed consultations in which eczema was often presented as a secondary problem that was dealt with rapidly as the appointment time came to a close. Others tried to avoid waste of perceived precious resources, 'I try not to waste an NHS appointment on eczema'.

A common lay belief was that practitioners in primary care had little expertise in eczema care and so over time consultations diminished, 'waste of time, wouldn't go back again'. Participants expressed frustration about consultations, particularly with GPs, describing a sense of being given the standard treatment using whichever product the practitioner happened to be familiar with: 'it looks like eczema here's some stuff', 'try this and come back' or 'I feel like it's just standard eczema … steroid, emollient off you go'. There was a perception of being prescribed the cheapest available product regardless of their needs. Some were frustrated by being offered conflicting information by practitioners. For example 'inconsistent care starting from the beginning and all the way through' and 'seeing a different GP all the time. … given different things. … it got so difficult' and 'one doctor emollients then steroids, another doctor steroids then emollients'.

Lay people reported that practitioners appeared to expect them to know what eczema is. None of the participants recalled being offered an explanation of the condition in primary care. Typical comments included,

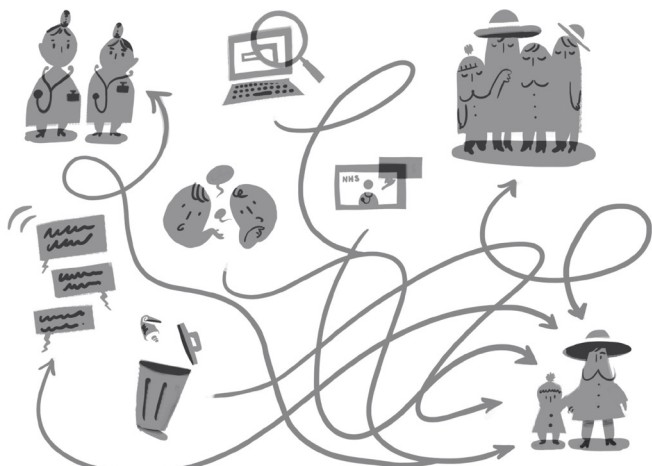

**Figure 2** Lay eczema mindlines. Sources of information underpinning eczema lay mindlines.

'expected me to know what eczema is' and 'just told me it was a dry skin condition'. Particular discontent was evoked by the lack of explanation that eczema may be a long-term condition that requires consistent management even when at its best, 'not told it's a long-term condition and will take time to control', 'they don't say, this is the reason this keeps happening …it's flare after flare' and 'wasn't any sense of it being a kind of care plan … very much ad hoc'. Advice was limited to applying emollients regularly and using the minimum amount of topical steroid for the shortest possible time. The advice to use emollients regularly was often taken to mean regularly for as long as the eczema was troublesome rather than on an ongoing basis. In many cases, this interpretation led to poor long-term control and reactive management.

### Theme 3: I need to manage this myself

Having experienced some of the challenges in the theme 'not worth consulting', lay people appeared to reach a trigger point at which they realised that they were going to have to take an active role in managing their, or more commonly their child's, eczema. This group constitutes those self-managing by default. They described a long and uncomfortable journey borne out of sheer exasperation with primary care services. One mother reported 'getting to a place of being able to live and deal with it takes such an extraordinary amount of time'. There was a sense of suffering; they had learnt how to manage eczema by default, often in the absence of information from practitioners, 'need more information earlier but seems it's up to the patient to get it'. They had 'learnt the hard way' and had developed knowledge about what works for them. The realisation that self-management was essential came in different ways with some experiencing an epiphany 'suddenly clicked … I had to manage it …. I wanted to find out the facts …… the brutal facts of everything'. Learning about eczema care increased their sense of control 'gave me lots of knowledge, I didn't have to act on it … but I know a bit more'.

Several participants reported that when they became more informed they experienced improved quality of care as practitioners were more likely to acknowledge their personal expertise. These encounters were highly valued and people adopted strategies to ensure that they were able to consult with a specific GP either by telephone or in person. They described a different type of consultation in which 'I felt listened to … it was a conversation' and 'my own GP remembered me …. she had a lot of ideas on how to treat it and so on'. Others presented alternative pictures of more proactive consultations for which they had prepared and in which they encountered shared decision making 'they do ask my opinion, decision making is 50:50'. Others described their discomfort at having to use more forceful approaches to get the care they knew from experience was required. They explained how 'I've learnt to go in there and be quite pushy …. *this* is what I need' and having to say 'I need this, don't give me anything else'. People were infuriated when repeat prescriptions were changed without discussion, 'you just get given whatever they've got, sometimes it's swapped without explanation'. Occasionally, self-management was thwarted by practitioners not acknowledging people's expertise. For example, one young man with frequent flares knew from experience that early treatment with a moderate potency steroid resulted in quick and effective control. However, the GP decided that his skin was not bad enough and so would only prescribe a low potency preparation. This led to a further consultation a few days later as his eczema deteriorated.

A small number of lay people reported a perceived need to be referred to secondary care but this tended to be a slow process requiring sustained effort by the individual. It often followed multiple primary care consultations and was sometimes the result of sheer determination. One mother reported how she had to 'force for a dermatology appointment … fight for that, put his case forward'.

### Theme 4: how I know what to do

Beliefs about eczema, and where responsibility for care lay, impacted the way in which lay mindlines were constructed. Those who were content for care to be directed by practitioners sought little knowledge other than that provided during consultations. Those who were content to self-manage and those who self-managed by default used similar strategies to gather knowledge but to a different intensity. The former group described 'dabbling' for interest rather than a purposeful effort to find out more. They also differed in the way in which they used the resultant knowledge. There did not seem to be a relationship between demographic characteristics and how knowledge was acquired. Parents of children with eczema were more likely than adults with the condition to invest significant time and effort in seeking out treatment possibilities. Many parents reported how difficult it was to watch their child suffer, and this drove the desire to access the best possible knowledge.

Sources of information underpinning lay mindlines were diverse (figure 2). Personal experience was a major factor with participants reporting 'I've tried everything … I know what works for me', 'I've worked it out through experience' and 'if I don't do it [apply emollients] I know I'll pay for it later'. Information from family and friends was an influential source that was transmitted and used in differing ways. Two participants came from families with multiple children with eczema; this resulted in shared knowledge and also some sharing of over-the-counter and prescribed medications. For example, 'trying things and testing things … we're a big family, we share knowledge' and 'in my family we kind of shared prescribed treatments'. Personal experience was often imparted among friends and acquaintances. Mothers of young children reported conversations at the school gates and during activities when their child's skin could be seen such as sports and swimming clubs. Mothers deemed these conversations helpful and did not take offence to another parent commenting on their child's eczema. Participants spoke of some 'old wives tales', knowledge they had assimilated from unknown sources and which was often perpetuated in informal encounters with others. The most frequently recounted topic of information related to the potential dangers of topical steroids. They described 'just knowing', 'steroids … back of your mind, bad for the skin, thins it, don't know where from'. 'Just knowing' was also alluded to by a mother of a child with severe eczema, 'instinctively I knew it was bad … I knew in my head that I had to get him to the hospital'.

The internet provided a wealth of information but, again, patterns of seeking and using this information differed. Participants universally focused on seeking treatment possibilities rather than better understanding the condition. They recounted two types of searches, often completed in parallel. Focused explorations were most frequently reported of the websites *NHS Choices* and *patient.co.uk*. These tended to be trusted sources of information, 'NHS website … really good … proper things … not just someone in the USA pretending to be a doctor'. However, for some, these websites were only a starting point, 'good as a basis'. Occasional participants recounted visiting these sites alongside their GP or HV during consultations. Only one participant reported visiting the condition specific National Eczema Society website.

Additionally, many participants described 'Googling' information. The motivations for a Google search varied from desultory internet wandering in a spare few minutes, often by 'content self-managers', to sustained and focused efforts described by 'self-managers by default'. Participants all expressed concern about the veracity of internet information, 'websites may be dodgy but they sort of make sense'. They had varying levels of confidence in their ability to judge quality of information. Some participants had high levels of ability to critique offerings and others were 'wary about interneting …. great deal of non-information, wouldn't trust myself to filter'. Online forums evoked a range of opinions from 'wary of forums, weird stuff, airy fairy' to a much more trusting perception, 'if it's worked for quite a lot of people I'll try' and 'get more real answers, they're the ones who know how it feels'. First-person accounts in the media and 'medical' television programmes could be powerful influences, for example, a BBC television programme described as a 'proper programme with proper doctor'. Advertising was largely seen as irrelevant.

Two significant factors were associated with lay mindline development. First, trusting the source was an important factor for many participants. The websites *NHS Choices* and *patient.co.uk* and information from other parents appeared to be the most trusted sources for many people. Second, realness concerned the extent to which the person imparting information actually understood eczema and the trials of living with this condition. If perceived as real, the believability of information was enhanced. Participants' ability to evaluate evidence from different sources varied from those who avoided some sources 'internet scary' to people who possessed all the skills required to understand complex research reports. However, there was a commonality in the extent to which they would put new information into action. Topical treatments were generally perceived as safe, and many would order these and use them without consultation or disclosure during subsequent consultations. Others would seek practitioner advice before use. All participants drew a line at using unprescribed oral medication.

## DISCUSSION

People with eczema and parents of children with eczema broadly fit one of four broad typologies of approach to self-management: content to self-manage; content to accept practitioner management; self-managing by default; and those referred to secondary care. Lay mindlines in eczema are a complex amalgamation of sources of knowledge and experience, which is gathered over time and influenced by perceived disease severity and burden. Content self-managers and those self-managing by default used similar sources of information but intensity of knowledge seeking and ways in which information was used were quite different. For some participants, mostly parents, there was a 'trigger point' at which they realised that they had to play an active role if long-term control was to be achieved. At this point, they were highly receptive to new information. Identifying and targeting people who are approaching this point would be an effective opportunity for influencing lay mindlines and promoting more equal and useful consultations and thus potentially more effective eczema self-management. This capturing of the right time to intervene is comparable with the 'teachable moment', which Lawson and Flocke[40] suggest can be created through effective practitioner-patient interactions. Findings from this study corroborate the notion that useful mindline development is predicated on good relationships and collaborative learning.[9]

This study is one of the first to apply mindline theory to lay people and offers new insights about how information is gathered and used. The findings identify a group of lay people who may be most receptive to new approaches to supporting self-management of eczema in primary care through interventions to amend mindlines. Ethnographic data have been collected in one general practice for depth and in a large and super diverse geographical area for breadth.[41] The study conforms with conventions of robust qualitative work. It is rigorous (coherent and sufficiently well reported to be open to external audit), relevant (enriches understanding of the subject), resonant (resonates with readers experiences and understandings) and reflexive (subjectivity of the author is acknowledged).[38] Limitations include issues of reliability as the ethnographer is a lone worker; however, this was mitigated by conversations with participants to check understandings. Equally, findings may not be transferable, but the breadth of data collection should minimise this risk.[42] Data on disease severity, which may have added to the findings, were not collected. It was not feasible to collect these data as they were not routinely captured in consultations, and it would not have been appropriate for the researcher to examine participants within the confines and focus of the study.

It is well recognised that living with eczema is difficult.[43 44] However, there is limited qualitative research about this experience. A recent scoping review identified only 22 studies in mainstream literature.[45] This study contributes to the body of knowledge adding nuanced understandings about how people approach self-managing eczema and the knowledge used to underpin this. Findings from this study on the challenges of self-management largely concur with existing literature. Practices are influenced by: carer's beliefs about eczema treatments; the time consuming nature of treatments; child resistance[46]; lack of knowledge, skills and confidence[47 48]; and difficulty in identifying reliable information from the vast available volume.[49 50] Observations on the desire for knowledge and greater control over the condition reaching a trigger point has similarities with a study of parents of children with other long-term conditions in which information seeking patterns change over time and according to where the person is in their life course.[51]

As with other studies, primary care consultations were often considered unsatisfactory by lay people (patients or parents) who commonly felt that the condition was being trivialised by practitioners[16 52 53] and who perceived that practitioners lacked knowledge.[50] The notion of a group who are self-managing by default has parallels with the involuntary autonomy described by Noerreslet and colleagues[54] in which patients thought they 'had to' be responsible for their care. This led to an unwanted level of independence when their preference was for a partnership with shared responsibility. Similarly, there is a dichotomy between patients feeling that they are offered choices and them actually wanting to make choices about healthcare.[55 56] Many participants in the current study desired genuine shared decision making, which remains challenging to achieve in general practice.[57]

Current adjunct eczema interventions are based on self-management or educational or psychological input designed to improve self-management capabilities and change health behaviours. The impact of such interventions is variable, active ingredients for success are not yet known and they are costly and have limited availability.[29–32] Current KM activity has predominantly focused on movement of knowledge among researchers and practitioners, although there is a growing drive towards active engagement of patients and the public.[5] To date, only one study has explicitly sought to understand patient mindlines with Oduro-Mensah and colleagues[10] using the term clientlines. No study has yet investigated how lay mindlines may best be revised or modified. There has been limited investigation into modification of practitioner mindlines,[9] which may provide a starting point for further layperson focused work.

This study identifies a distinct group, people self-managing eczema by default, who have reached a trigger point at which they are desperate for reliable knowledge about eczema care. They desire consultations with well-informed practitioners within which their personal experience and tacit knowledge is valued and integrated into agreed plans for long-term control. While current adjuncts to eczema care have a place they also have limitations as discussed earlier. This study offers a new approach to supporting self-management through active amendment of lay eczema mindlines that may be feasible during routine consultations. Consistency of care and enduring relationships are key to successful eczema treatment, and for these reasons, nurses with requisite knowledge and skills may be best placed to provide cost-effective ongoing care.[58–60]

## CONCLUSION

This ethnographic study provides evidence that lay eczema mindlines exist and that they share commonalties in how they are developed. This finding offers a new approach to lay-focused KM. The outstanding challenge is to find strategies to revise or modify these mindlines by adding reliable and useful knowledge and by erasing outdated or inaccurate information. In the case of eczema, it is worth considering in more detail how the universal caution about use of topical steroids has permeated at all levels and use this approach to amend lay mindlines. Addressing lay mindlines has the potential to contribute to more effective eczema self-management in primary care particularly among people who are identified as being most receptive.

**Acknowledgements** Thanks to Amanda Roberts, who have given invaluable lay feedback on the planning and design of this study. Thanks to James Mycock, Birmingham City University for the mindline illustration and to Dr Andrew Booth, University of Sheffield, for his valuable feedback on earlier iterations of this manuscript.

**Contributors** FC is the sole contributor to this paper.

**Funding** This report is independent research arising from a Knowledge Mobilisation Research Fellowship, Professor Fiona Cowdell, KMRF-2015-04-004, supported by the National Institute for Health Research.

**Disclaimer** The views expressed in this publication are those of the authors and not necessarily those of the NHS, the National Institute for Health Research, Health Education England or the Department of Health.

**Competing interests** None declared.

**Patient consent** Obtained.

**Ethics approval** The study was approved by a University Ethics Committee and HRA Yorkshire & The Humber – Leeds West Research Ethics Committee (ref no:16/YH/0252).

**Provenance and peer review** Not commissioned; externally peer reviewed.

**Data sharing statement** The datasets generated and/or analysed during the current study are not publicly available as they are not designed to be reanalysed by others but are available from the corresponding author on reasonable request.

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
