## [Reviewer comments · BMJ Open]

ARTICLE DETAILS

TITLE (PROVISIONAL)	Knowledge mobilisation: an ethnographic study of the influence of lay mindlines on eczema self-management in primary care in the United Kingdom
AUTHORS	Cowdell, Fiona

VERSION 1 – REVIEW

REVIEWER	Matthew Ridd University of Bristol, UK
REVIEW RETURNED	05-Feb-2018

GENERAL COMMENTS	Thank you for asking to me read this well conducted and reported study with interesting findings. I think it's suitable for publication as submitted, and have the following minor comments/suggestions: 1. Table 2: would benefit from an "identifier" column, with this identifier cited whenever a quote is used so that the reader better understands the characteristics of the participant quoted; and "eczema severity" and "confidence with self-care" (e.g. low, medium or high would do) columns ideally, although I appreciate that if these data were not collected, this will not be possible – "how bothersome" might be a good alternative for eczema severity.2. Related to point 3, I note the author in the discussion writes: "Data on disease severity, which may have added to the findings, was not collected. It was not feasible to collect this data as it was not routinely captured in consultations and it would not have been appropriate for the researcher to examine participants within the confines and focus of the study." The arguments made for not collecting this are reasonable, but participants could have been asked to report a global self-assessment of severity (mild, moderate or severe) – in the same way that one question on the topic guide asks how bothersome the eczema is.3. I wasn't clear whether the author set-out to construct the typographies described, or whether these emerged in an inductive way as the themes did?4. Page 8: "P 8: Each theme is discussed with examples from the data below. Lay person typologies and themes are cross-cutting although the focus in this paper is on the first two levels are it is these groups who are wholly treated in primary care." I found this sentence ambiguous – is the author referring to the first two typologies or themes here?5. Page 9: "Knowledge was predominantly from clinicians, with nurses often reported as providing the most useful information particularly imparting "the simple stuff" such as about how to use topical treatments." I was surprised to read that nurses are imparting information on topical treatments because this is not common place in primary care. This is something possibly unique to the GP surgery that she studied, or participants may have been referring to
---

	community dermatology or dermatology nurse specialists seen in secondary care? 6. Page 13: "Parents of children with eczema were more likely than adults with the condition to invest significant time and effort in seeking out treatment possibilities." I suspect this may be because adults learnt/tried this when a child? Or does the author really think this is a difference born of the age of participants, rather than the "stage of their disease career"? 7. I was surprised that there was no mention of concerns about (food) allergy or allergy testing. Did this not arise in discussions/observations or has the author chosen not to report this aspect? (Which may act as a barrier to self-management.) 8. More generally, some of the sentences are long – this might be a stylistic issue, but I think for an international audience/reader with English as a second language, the readability could be improved in places by using splitting some sentences into two. 9. Similarly, there are minor punctuation errors, such as a missing possessive apostrophe.
--	---

REVIEWER	Parker Magin University of Newcastle, Australia
REVIEW RETURNED	18-Mar-2018

GENERAL COMMENTS	This manuscript reports a study which takes an approach to the interpretation of patients' self-management of eczema by examining patients' thinking about their condition via the lens of mindlines. As well as this interesting and novel approach, there is interesting data presented. This impact, however, is attenuated by a number of issues in the structure of the study and presentation of the manuscript. The rationale for the combination of ethnographic observation of unselected patients in a single practice and the interviews with patients/parents of patients with eczema is not established and the results of the observation presented are limited in extent, quite thin, and not well-integrated with the interview results. It is hard to see what the observational material adds to the manuscript. It could be imagined that an understanding (via the ethnographic approach) of the general practice environment in which the patients experience healthcare could frame their experience of eczema, but there is no strong sense in this manuscript of that being the case. And it could be equally argued that, with a vast literature on the patient experience in UK general practice, the observations in a single practice here could do little to further the understanding of the context of the patients' experience of their condition. The mindlines lens really only seems to be operating in the fourth of the four themes, 'how I know what to do'. The data in the first three themes is interesting and is contextual for the fourth theme but is pretty much consistent with previous qualitative studies' findings in this area. So more emphasis could be given to presenting material related to the fourth theme and its relationship to mindlines. It's stated that the lay person typologies are 'cross-cutting' with the themes, but little is actually presented in the findings regarding these. Especially as they are referred to in the Discussion as being of importance, data supporting the formulation of the typologies should be presented. Other specific points are: The recruitment of participants for the interviews needs to be better reported. It is stated that participants were recruited from the practice at which the ethnographic observations were conducted plus 'a wider super-diverse geographical area' – what is a super-
---

	diverse geographical area and how many participants were from this and how many were from the single practice? How were these participants/practices identified and invited to participate? It's stated that 'interviewees were recruited via invitation letter', but what was the sample frame for this recruitment and how was it constructed? How were the parameters on which maximum variation sampling was conducted elicited? It's stated that clinical and lay colleagues corroborated interpretations. How could they corroborate findings if not part of the research team with a familiarity with the data? It would be useful for the quotes in the Results section to be labelled to allow orientation to the demographic data on each participant presented in Table 2.
--	--

VERSION 1 – AUTHOR RESPONSE

Thank you for the valuable comments, I have addressed each below

Editor comments	Response
Please reduce the number of identifiers in table 2, of patient demographics, to ensure anonymity of the participants. We normally recommend including a maximum of two identifiers.	I have reduced identifiers to name and age only in Table 2
Please include the study design and setting/country in the title.	I have amended title to: Knowledge mobilisation: an ethnographic study of the influence of lay mindlines on eczema self-management in primary care in the United Kingdom
Please re-upload FIGURE with at least 300 dpi resolution.	Both changed to 300dpi and uploaded
Reviewer 1 comments	
Table 2: would benefit from an "identifier" column, with this identifier cited whenever a quote is used so that the reader better understands the characteristics of the participant quoted; and "eczema severity" and "confidence with self-care" (e.g. low, medium or high would do) columns ideally, although I appreciate that if these data were not collected, this will not be possible – "how bothersome" might be a good alternative for eczema severity.	Thank you for this comment. I have reduced the number of identifiers to two. I can see that this may be useful information but would lead to a high risk of identification.
Related to point 3, I note the author in the discussion writes: "Data on disease severity, which may have added to the findings, was not collected. It was not feasible to collect this data as it was not routinely captured in consultations and it would not have been appropriate for the researcher to examine participants within the confines and focus of the study." The arguments made for not collecting this are reasonable, but participants could have been asked to report a global self-assessment of severity (mild, moderate or severe) – in the same way that one question on the topic guide asks how bothersome the eczema is.	I agree that I could have asked about disease severity, but the focus on my work was on mindlines.
I wasn't clear whether the author set-out to construct the typographies described, or whether these emerged in an inductive way as	They emerged inductively, I have added to the manuscript on page 8

the themes did?	
Page 8: “P 8: Each theme is discussed with examples from the data below. Lay person typologies and themes are cross-cutting although the focus in this paper is on the first two levels are it is these groups who are wholly treated in primary care.” I found this sentence ambiguous – is the author referring to the first two typologies or themes here?	I have amended to “Each theme is discussed with examples from the data below. Lay person typologies and themes are cross-cutting although the focus in this paper is on the first two typologies are it is these groups who are wholly treated in primary care.”
Page 9: “Knowledge was predominantly from clinicians, with nurses often reported as providing the most useful information particularly imparting “the simple stuff” such as about how to use topical treatments.” I was surprised to read that nurses are imparting information on topical treatments because this is not common place in primary care. This is something possibly unique to the GP surgery that she studied, or participants may have been referring to community dermatology or dermatology nurse specialists seen in secondary care?	Good point, I have amended to: “Knowledge was predominantly from clinicians. Eczema consultations with nurses were rare but reported as providing the most useful information particularly imparting “the simple stuff” such as about how to use topical treatments.”
Page 13: “Parents of children with eczema were more likely than adults with the condition to invest significant time and effort in seeking out treatment possibilities.” I suspect this may be because adults learnt/tried this when a child? Or does the author really think this is a difference born of the age of participants, rather than the “stage of their disease career”?	The point here is that parents would seek information about their child’s health much more actively than adults with the condition. The implication is that It is much harder to watch your child suffer than to suffer yourself. I’ve added Many parents reported how difficult it was to watch their child suffer and this drove the desire to access the best possible knowledge.
I was surprised that there was no mention of concerns about (food) allergy or allergy testing. Did this not arise in discussions/observations or has the author chosen not to report this aspect? (Which may act as a barrier to self-management.)	Food was mentioned occasionally. As the focus was on mindlines my interest was in the way that knowledge was accrued rather than knowledge content.
More generally, some of the sentences are long – this might be a stylistic issue, but I think for an international audience/reader with English as a second language, the readability could be improved in places by using splitting some sentences into two.	I have divided some longer sentences in the manuscript.
Similarly, there are minor punctuation errors, such as a missing possessive apostrophe.	I have corrected minor punctuation errors
Reviewer 2	Thank you for the positive comments
The rationale for the combination of ethnographic observation of unselected patients in a single practice and the interviews with patients/parents of patients with eczema is not established and the results of the observation presented are limited in extent, quite thin, and not well-integrated with the interview results.	The rational for choice is offered in this sentence on page 16 “Ethnographic data has been collected in one general practice for depth and in a large and super-diverse geographical area for breadth (41)”. I have added this earlier I the manuscript as well for clarity. All data is an amalgamation of observation and interviews. I have chosen not to separate as in this ethnographic study I am aiming to present a rich and holistic insight using all data gathered.
It is hard to see what the observational material adds to the manuscript. It could be imagined that an understanding (via the ethnographic	As above, the observational data are integrated throughout – for example, I observed the frustration of patients telling me how difficult it is

approach) of the general practice environment in which the patients experience healthcare could frame their experience of eczema, but there is no strong sense in this manuscript of that being the case. And it could be equally argued that, with a vast literature on the patient experience in UK general practice, the observations in a single practice here could do little to further the understanding of the context of the patients' experience of their condition.	to get an appointment. I saw patients bringing up eczema at the very end of the consultation. I witnessed emollients being prescribed with no information about how to use them effectively.
The mindlines lens really only seems to be operating in the fourth of the four themes, 'how I know what to do'. The data in the first three themes is interesting and is contextual for the fourth theme but is pretty much consistent with previous qualitative studies' findings in this area. So more emphasis could be given to presenting material related to the fourth theme and its relationship to mindlines.	Thank you, this comment suggests that I have made the point effectively. Mindlines are context-specific and so to understand them there is a need to look at the bigger picture which I have offered in earlier sections of the manuscript.
It's stated that the lay person typologies are 'cross-cutting' with the themes, but little is actually presented in the findings regarding these. Especially as they are referred to in the Discussion as being of importance, data supporting the formulation of the typologies should be presented.	Thank you, this is a really useful point and I've added further detail on page 9
The recruitment of participants for the interviews needs to be better reported. It is stated that participants were recruited from the practice at which the ethnographic observations were conducted plus 'a wider super-diverse geographical area' – what is a super-diverse geographical area and how many participants were from this and how many were from the single practice?	Detail added on page 6 As previously my aim is to provide a rich and holistic insight using all data gathered rather than focus on individuals
How were these participants/practices identified and invited to participate? It's stated that 'interviewees were recruited via invitation letter', but what was the sample frame for this recruitment and how was it constructed? How were the parameters on which maximum variation sampling was conducted elicited?	Thank you, my omission, detail added on page 6
It's stated that clinical and lay colleagues corroborated interpretations. How could they corroborate findings if not part of the research team with a familiarity with the data?	Good point, perhaps I have changed the term corroborated on page 8
It would be useful for the quotes in the Results section to be labelled to allow orientation to the demographic data on each participant presented in Table 2.	Having reduced the number of identifiers to two I do think that this would add to the manuscript

I have changed the term clinician to practitioner throughout as this term better reflects that the study is concerned with all health care practitioners not just doctors.

VERSION 2 – REVIEW

REVIEWER	Parker Magin University of Newcastle, Australia
-----------------	--

REVIEW RETURNED	17-Apr-2018
-------------

GENERAL COMMENTS	A number of comments from the original review have not been addressed. The description of recruitment of participants is still not adequate: 'via an institutional website' provides the reader with no indication of from where these participants were recruited or what kind of individuals they may be. The transferability of the results is not clear. Also, 'people with medically diagnosed eczema' isn't a sample frame. The manuscript needs to say how people were chosen to send invitation letters to. It is still not clear what constituted the 'wider super-diverse geographical area'. And how is it assessed that it encompassed 'citizens of multiple nationalities and also diversity of legal statuses, of socio-economic conditions and a greater range in how people choose to live and define themselves'? How many participants were recruited from the single practice and how many from the second source population? I don't think that non-members of the research team, without a knowledge of the study data, 'confirming the resonance of the findings' is contributory to the rigour of the study. I think this should be omitted. The gender and, especially the age, of patients with eczema are important contextual factors for the nature of the condition and for responses to it and for interactions with medical practitioners and nurses
---

VERSION 2 – AUTHOR RESPONSE

Thank you for the valuable comments, I have addressed each below

11.6.18

Reviewer comment	Response
The description of recruitment of participants is still not adequate: 'via an institutional website' provides the reader with no indication of from where these participants were recruited or what kind of individuals they may be.	Further detail added via a higher education institute website with a reach of over 5000 people including staff, both academic and professional services, and external subscribers. Several participants were recruited by word of mouth from people who had seen the web recruitment information.
The transferability of the results is not clear.	The responses below give evidence of the transferability of this work
Also, 'people with medically diagnosed eczema' isn't a sample frame. The manuscript needs to say how people were chosen to send invitation letters to.	Firstly invitation letters were sent to patients from the General Practice who had a diagnosis of eczema recorded in their medical records and who had been prescribed emollients during the last year, indicating that their eczema may be a concern. Letters were sent to a group selected to represent the broad spectrum of patients including different age, gender and nationality.
How many participants were recruited from the single practice and how many from the second source population?	Participants were lay people registered at the observed practice (n=8) and lay people recruited via the website (n=8). Lay participants were all residents in the Midlands of the United Kingdom which is recognised as being an area of super-diversity, defined as an area with citizens of
It is still not clear what constituted the 'wider super-diverse geographical area'. And how is it	

assessed that it encompassed 'citizens of multiple nationalities and also diversity of legal statuses, of socio-economic conditions and a greater range in how people choose to live and define themselves'?	multiple nationalities and socioeconomic status.
I don't think that non-members of the research team, without a knowledge of the study data, 'confirming the resonance of the findings' is contributory to the rigour of the study. I think this should be omitted.	I have removed this sentence
The gender and, especially the age, of patients with eczema are important contextual factors for the nature of the condition and for responses to it and for interactions with medical practitioners and nurses.	I agree this is important contextual data. I have provided a table of interview demographics (age and gender). My intention in this manuscript is to give insight into a vast amount of data. The data revealed no patterns that were age / gender specific.